# MiRNA as a Potential Target for Multiple Myeloma Therapy–Current Knowledge and Perspectives

**DOI:** 10.3390/jpm12091428

**Published:** 2022-08-31

**Authors:** Aneta Szudy-Szczyrek, Sean Ahern, Janusz Krawczyk, Michał Szczyrek, Marek Hus

**Affiliations:** 1Chair and Department of Haematooncology and Bone Marrow Transplantation, Medical University of Lublin, 20-081 Lublin, Poland; 2Department of Haematology, University Hospital Galway, H91 Galway, Ireland; 3National University of Ireland, H91 Galway, Ireland; 4Chair and Department of Pneumonology, Oncology and Allergology, Medical University of Lublin, 20-950 Lublin, Poland

**Keywords:** miRNA, multiple myeloma, oncomirs, tumor suppressor, biomarkers

## Abstract

Multiple myeloma (MM) is the second most common hematological malignancy. Despite the huge therapeutic progress thanks to the introduction of novel therapies, MM remains an incurable disease. Extensive research is currently ongoing to find new options. MicroRNAs (miRNAs) are small, non-coding RNA molecules that regulate gene expression at a post-transcriptional level. Aberrant expression of miRNAs in MM is common. Depending on their role in MM development, miRNAs have been reported as oncogenes and tumor suppressors. It was demonstrated that specific miRNA alterations using miRNA mimics or antagomirs can normalize the gene regulatory network and signaling pathways in the microenvironment and MM cells. These properties make miRNAs attractive targets in anti-myeloma therapy. However, only a few miRNA-based drugs have been entered into clinical trials. In this review, we discuss the role of the miRNAs in the pathogenesis of MM, their current status in preclinical/clinical trials, and the mechanisms by which miRNAs can theoretically achieve therapeutic benefit in MM treatment.

## 1. Introduction

Multiple myeloma (MM) is a malignancy of plasma cells that accounts for 1.8% of all cancer cases [1]. Worldwide, an estimated 176,404 people were diagnosed with MM in 2020. MM is classically considered as a disease of older adults; the median age at diagnosis is 70 years [2,3]. The disease can cause failure of the bone marrow, leading to anemia, destructive bone lesions, kidney failure, immune disorders with recurrent infections, and hypercalcemia. In recent years, the increase in the number of therapeutic options, such as proteasome inhibitors, immunomodulatory drugs, and monoclonal antibodies, has significantly changed the prognosis for MM, leading to an increased length of survival. A 5-year survival rate reported in the SEER database is estimated at 57.9% (2012–2018) [4].

The development of MM is controlled by both genetic and epigenetic mechanisms. Genome instability is an early event in the development of the disease and a hallmark of patients with monoclonal gammopathy of undefined significance (MGUS)—a precancerous condition preceding MM. The IGH-mediated translocation and hyperdiploidy (trisomy) are two distinct molecular pathways that initiate neoplastic trans-formation and are key pathways to the pathogenesis. As the disease progresses, MM is characterized by increasing and often extreme genetic abnormalities [5,6]. A number of epigenetic abnormalities have also been noted in patients with MM, the importance of which is still under investigation. The epigenetic effect is defined as changes in gene expression resulting from modification of the chromatin structure without changes in the DNA sequence. Epigenetics includes mechanisms such as DNA methylation, histone modifications: acetylation, methylation, phosphorylation, ubiquitination and sumoylation, and non-coding RNAs, i.e., microRNA (miRNA), long non-coding RNA (lncRNA), small interfering RNA (siRNA), and RNA interacting with piwi (piRNA) [7,8].

## 2. MicroRNA

MicroRNAs (miRNAs) are a class of short (about 22 nucleotides in length) non-coding RNA fragments that regulate post-transcriptional gene expression. They are mostly transcribed by RNA polymerase II (Pol II), which generates “hairpin” structure precursors known as pri-miRNAs [9]. The pri-miRNAs are cleaved by Drosha, a specific double-stranded RNA endoribonuclease, and a stem-loop structure of ~70 base pairs long (pre-miRNAs) is produced [10,11]. Pre-miRNAs are actively transported by exportin 5 to the cytoplasm where they are cleaved by another RNase III type endonuclease, Dicer1, to generate a ~20-nucleotide miRNA duplex. As a result, molecules without a loop connecting the 3′ and 5′ arms are formed, creating a double-stranded form—duplex miRNAs (miRNAs-3p and miRNAs-5p). The mature miRNAs are selectively bound together with the Argonaute AGO2 protein to the RNA-induced silencing complex (RISC), which plays a crucial role in silencing the expression of specific genes in the process of RNA interference. By binding to the 3′-UTR region of the target mRNA, miRNAs cause translational repression or destabilization of the mRNA. In the case of full complementarity with the 3′-UTR region, the mRNA is cut. Other miRNAs have the ability to inhibit translation by binding to the 5′-UTR region or to the RNA region constituting the open reading frame [9,12].

Genes for miRNAs are often located at fragile sites on chromosomes [13]. A common change in gene expression for miRNAs is observed in tumor cells, caused by, e.g., deletions, amplifications, or translocations. These alterations lead to changes in the expression of target genes. Depending on which genes they influence, miRNAs can function as oncomirs-procarcinogenic or as suppressors-inhibiting oncogenes. Therefore, miRNAs can be promising biomarkers in the diagnosis, prognosis, and treatment of cancer [14,15,16].

The disturbances in miRNA expression are closely related to MM development, and miRNAs seem to be an attractive research area for new therapeutic targets in MM [17,18,19].

## 3. Oncogenic and Tumor Suppressor miRNAs in MM

So far, a number of miRNAs with oncogenic potential have been identified, the overexpression of which is associated with the development or progression of MM. On the other hand, many studies have observed decreased expression of miRNAs with suppressor functions that act to inhibit oncogenes and reduce tumor growth. Functional studies have elucidated, at least partially, the mechanisms by which selected molecules can promote MM cell growth and expansion (Figure 1).

### 3.1. The Role of miRNAs in the Bone Marrow Microenvironment

Specific interactions between microenvironmental cells (in particular endothelial cells and bone marrow stromal cells) and the tumor cell clone determine MM growth, proliferation, and expansion [20,21]. Recent discoveries in cancer biology have revealed that BMSC-derived extracellular vesicles (EVs) are of key importance in communication [22,23]. EVs contain proteins, lipids, DNA, messenger RNA, miRNA, and long non-coding RNAs that are transported between selected cells [24,25,26]. miRNAs involved in the regulation of vascular development, reprogramming of fibroblasts and T lymphocytes have all been reported.

miR-10a is overexpressed in EV while intracellular expression in MM-BMSC is decreased, suggesting that miR-10a is actively released into the extracellular matrix. The inhibition of EV release causes accumulation of intracellular miR-10a, inhibition of cell proliferation and induction of apoptosis in MM-BMSC. Moreover, miR-10a derived from MM-BMSCs transferred to MM cells via EV enhances their proliferation [27]. It has been shown that a potential miR-10a target gene is *EphA8*, encoding the ephrin receptor. High *EphA4* expression promotes proliferation and drug resistance mediated by cellular adhesion associated with activation of the AKT pathway [28]. *EphA8* may be involved in MM progression by regulating the expression of an axon guidance molecule, SEMA5A. SEMA5A was also identified to be highly expressed in MM patients and led to their decreased survival time [29].

Overexpression of miR-27b-3p and miR-214-3p triggers proliferation and apoptosis resistance in MM fibroblasts via the FBXW7 and PTEN/AKT/GSK3 pathways, respectively. It has been proven that neoplastic plasmocytes, by releasing exosomes containing WWC2 protein, increase the expression of both miR-27b-3p and miR-214-3p in fibroblasts, which causes their reprogramming and consequently accelerates the transformation of MGUS to MM [30].

It has been shown that inhibition of miR-21 expression in naive T cells suppresses production of IL-17, which is an essential cytokine promoting MM progression and osteolysis through osteoclast activity. The main mechanism is the increase in the regulation of STAT-1/-5a-5b and the impairment of the STAT3 signaling pathways [31].

It has been demonstrated that the expression of miR-15a and miR-16 is significantly decreased in both MM cells and MM cell lines. Moreover, the expression of miR-15a/16 is inversely correlated with the expression of the vascular endothelial growth factor (VEGF-A). Ectopic overexpression of miR-15a/16 led to a decrease in the pro-angiogenic activity of MM cells. Finally, transfection with lentivirus-miR-15a or lentivirus-miR-16 results in significant inhibition of tumor growth and angiogenesis in mice [32].

It has been shown that miR-199a-5p leads to downregulation of several angiogenic factors such as VEGF-A, fibroblast growth factor (FGF-b), hypoxia-induced factors (HIF-1α), and IL-8. Furthermore, miR-199a-5p regulates MM spread processes. It has been shown that miR-199a-5p inhibits tumorigenesis through weakening malignant plasma cells’ chemotaxis [33,34].

### 3.2. The Influence of miRNAs on the Proliferation and Growth Processes of MM Cells

The miR-17-92 cluster located on chromosome 13q31.3 includes miR-18a, miR-20a, miR-92, miR-17, and miR-19a/b, and is activated by the proto-oncogene *MYC* and the transcription factor BHLH (C-MYC). Increased expression of miR-17-92 has been observed in various neoplasms [35]. It has also been shown that abnormally increased expression of the miR-17-92 cluster is involved in the malignant progression of MM [36]. The miR-17-92 cluster was upregulated in malignant MM cells as compared to normal plasmocytes [37]. Moreover, the miR-19a and miR-19b components reduced the expression of suppressor of cytokine signaling 1 protein (SOCS1), thereby accelerating the proliferation of MM cells. SOCS1 is a negative regulator of IL6-mediated signaling. It is suggested that decreased SOCS1 expression may induce signal transducer and activator of transcription 3 (STAT3) phosphorylation, ultimately resulting in unrestricted tumor cell growth. Moreover, it was revealed that miR-19 targets the *BCL2* gene and lowers the expression of the encoded protein, resulting in decreased apoptosis and increased proliferation of malignant PCs [38].

miR-21 is overexpressed in most cancer types and acts as an oncogene to target genes involved in proliferation, apoptosis, and metastasis. The effect of forced expression of synthetic miR-21 molecules on MM cells was assessed. It has been found that miR-21 exerted MM growth-promoting activity. Overexpression of miR-21 decreased the expression of *PTEN*, *BTG2*, and *Rho-B* mRNA. Furthermore, Western blot analysis showed that PTEN protein levels were decreased in MM cells [39].

miR-221 is another known oncogenic miRNA. The overexpression of miR-221 has been reported in many human cancers. The observed up-regulation of miR-221/222 in MM suppresses the expression of suppressor genes, and its reduction causes the inhibition of tumor cell growth both in vitro and in vivo. The canonic miR-221/222 targets include the following pathways: PTEN, PUMA, p27Kip1, and p57Kip2 [40].

The miR-29 family, including miR-29a, miR-29b, and miR-29c, has an inhibitory effect on tumor growth and is down-regulated in hematological neoplasms [41,42]. miR-29 is a prototypical example of epi-miRNA by targeting epigenetic regulators including DNA methyltransferases (DNMTs). MM discloses that the miR-29b target is DNMT, demethylating the SOCS1 gene and increasing its protein expression. Moreover, miR-29b can inhibit the PI3K/AKT signaling pathway, the AKT phosphorylation process and the P1 Forkhead box protein (FOXP1) pathway, thereby increasing the expression of apoptosis-promoting proteins (including P53 and caspase-9) and accelerating proliferation [43]. Wang et al. noted a significantly decreased expression of miR-29b in MM cell lines. MiR-29b downregulation was closely correlated with the International System Staging (ISS) stage. Exogenous overexpression of miR-29b, on the other hand, inhibited MM cell proliferation, induced cell cycle arrest, and induced apoptosis [44].

Downregulation of miR-26a is observed in MM patients compared to healthy volunteers. Hu et al. showed that induced overexpression of miR-26a reduces proliferation and migration and induces apoptosis in MM cell lines, and the CD38 protein is a direct target of miR-26a [45].

MM cell lines are characterized by decreased expression of miR-489. miR-489 acts as a tumor suppressor gene, inhibiting the viability and proliferation of malignant plasma cells. Additionally, miR-489 reduces glucose uptake and therefore ATP production. A potential target of miR-489 is lactate dehydrogenase-A (LDHA) [46].

Downregulation of miR-30-5p is also a common pathogenetic event in MM. It has been shown that this is the result of an interaction between MM cells and bone marrow mesenchymal stromal cells (BMSCs), which in turn increases the expression of *BCL9*, a transcriptional co-activator of the Wnt signaling pathway known to promote MM cell proliferation, survival, migration, and drug resistance [47]. It has been shown that the BMSCs of patients with MM are different in multiple respects to those of healthy patients and that they contain a lower amount of miR-15a in particular, which has tumor suppressive properties in MM cells and also in BMSCs [48]. Crosstalk between MM and BMSC cells occurs via miR-146a, which allows MM-derived exosomes with high miR-146a expression to be transferred into BMSCs, which then secrete cytokines and chemokines that sustain and enhance MM cell migration [49].

Next-generation sequencing has led to better resolution of small RNA expression and a study of bone marrow aspirates from 30 newly diagnosed MM patients using these methods has revealed the presence of a wider expanse of miRNAs, including miRNA-offset RNAs (moRNAs) than was previously reported in MM. This resulted in the annotation of 17 new miRNAs as well as the observation of 74 moRNAs with differential expression of moRNAs in MM subgroups [50].

### 3.3. The Role of miRNAs in the Mechanisms of Apoptosis and MM Cell Migration

The P53 protein plays a key role in regulating cell proliferation, mainly by inducing cell cycle endpoint, apoptosis or activation of DNA repair systems [51]. It has been proposed that miR-181a/b plays an important role in the regulation of P53. miR-181a/b has been reported to be elevated in MGUS and MM tumor plasmocytes [36,52]. It has been shown that miR-181a/b can negatively regulate P-300-CBP related factor (PCAF) expression, antagonize the positive effect of PCAF on P53, and ultimately decrease P53 expression. Additionally, miR-181a/b may act as a histone acetyltransferase to maintain P53 proteins at low concentrations or partially inactivate them [36]. In addition, miR-125a-5p, miR-194-2/192, and miR-215/194-1 are involved in the regulation of the p53 pathway in the course of MM [53,54].

MGUS and MM overexpress miR-106b. Several studies have demonstrated that the miR-106b/25 cluster is involved in cancer-related pro-survival/anti-apoptotic signaling pathways, including, e.g., the mitogen-activated protein kinase (MAPK) pathway [55]. It has been observed that the inhibition of miR-106b/25 leads to a suppression of the p38 MAPK expression and hence to a reduction in viability and induction of cell apoptosis [56].

miR-214-3p increases resistance to apoptosis in myeloma fibroblasts by targeting the apoptotic pathways of FBXW7 and PTEN/AKT/GSK3 [30].

miR-19b and miR-20a are other oncomiRs upregulated in MM cells. miR-19b/20a enhances plasma cell proliferation and migration as well as inhibits apoptosis. It has been shown that transfection of miR-19b/20a lowered the concentration of PTEN protein. Lentivirus-mediated delivery of miR-20a significantly accelerated tumor growth [57].

miR-27 is also overexpressed in MM and correlates with shorter overall survival. It has been noted that it promotes MM growth by enhancing cell proliferation, migration, and invasion through the Sprout 2 (SPRY2) homologue. Meanwhile, anti-miR-27 has the opposite effect. The inhibitors of miR-27 exert an anti-tumor effect on MM cells [58].

On the other hand, miR-15a and miR-16-1 downregulation are observed in MM cells. Both miRNAs were found to reduce MAPK signaling, AKT kinase, NF-κB-activator MAP3KIP3, and S6 ribosomal protein. Interestingly, transfection of miR-15a/16-1 suppressed growth and apoptosis in neoplastic plasmocytes [59].

miR-34a acts as a potent tumor suppressor and its expression is dysregulated and downregulated in a variety of cancers, including MM. By targeting stemness factors such as *NOTCH*, *MYC*, *BCL-2*, and CD44, miR-34a epigenetically negatively regulates cancer stem cells [60]. In animal models, miR-34a analogs suppress MM growth by activating apoptosis and inhibiting pro-survival signaling through the kinases CDK6, BCL2, and NOTCH1. Moreover, miR-34a has been demonstrated to reduce plasmocytes’ proliferation by inhibiting transforming growth interaction factor 2 (TGIF2) [61,62]. Decreased expression of miR-34a-5p in MM cell lines is associated with overexpression of mitochondrial RNA processing endoribonuclease (RMRP), which enhances cell proliferation. In turn, RMRP knockdown induces their apoptosis [63].

Suppressor functions are also displayed by miR-125a and miR-125b. MiR-125a reduces MM cell viability and colony formation capacity. There is evidence of decreased expression of miR-125a in MM cell lines. Ubiquitin specific peptidase 5 (USP5) was identified as a target for miR-125a. USP5 enhances cellular deubiquitination and proteolysis. Wu et al. reported that up-regulation of miR-125a and low expression of USP5 significantly inhibited MM tumor growth in vivo [64]. It has been shown that miR-125b is downregulated in MM patients by the tumor necrosis factor (TNF) and insulin-like growth factor (IGF-1) [65]. High expression of miR-125b inhibits tumor plasmocytes by inhibiting IRF4, which is critical for MM cell survival. Interestingly, miR-125b has been demonstrated to increase miR-34a levels, which in turn inhibits the IL-6/STAT3/miR-34a receptor feedback loop. The activation of these pathways results in MM cell death [66].

It has been reported that patients with MM show a constitutively low expression of miR-33b. Upregulation of miR-33b reduces MM cell viability, migration, and colony formation and causes increased apoptosis. PIM-1 kinase is a target for miR-33b, and it blocks the binding between the BCL2 agonist (Bad) and Bcl2/1-xl to inhibit apoptosis [67].

miR-155 is another molecule with a potential suppressor function in patients with MM. An oncogenic role for miR-155 was first reported in the context of Waldenstrom Macroglobulinemia [68]. It is significantly reduced in MM cells. It has been shown that miR-155 acts by inhibiting the proteasome, targeting the PSMβ5 subunit, and replacing miR-155 has antiproliferative and pro-apoptotic effects in the MM cell line [69].

miR-29b is responsible for the reduction of the expression of genes involved in proliferation and inhibition of apoptosis. Forced expression of miR-29b in MM cell lines inhibits cell growth and triggers apoptosis in vitro. Moreover, the ability of miR-29b to induce apoptosis in vivo was also demonstrated in an animal model of MM. miR-29b negatively regulates the expression of the Sp1 transcription factor in MM cells [70].

It has been observed that the expression of miR-101-3p in MM cells decreased, while the expression of survivin (BIRC5), a protein with an anti-apoptotic effect, was high. It has been confirmed that miR-101-3p, via survivin, reduces the viability of malignant plasma cells [71].

It has been demonstrated the potential of miR-137/197 as tumor suppressors is involved in the regulation of MM cell apoptosis by targeting *MCL-1*. The expression of miR-137/197 is significantly lower in MM cell lines and samples from MM patients compared to normal plasmocytes. miR-137/197 transfection resulted in decreased expression of the MCL-1 protein as well as induction of apoptosis, inhibition of viability, colony formation, and migration of MM cells. MCL-1 has been identified as a direct target of miR-137/197 [72].

Lists of selected clinically significant oncomirs and tumor suppressor miRNAs in MM are presented at Table 1 and Table 2.

## 4. miRNAs as Prognostic and Predictive Biomarkers in MM

Perhaps the best clinical use of miRNAs in MM lies in the realm of prognostication. Unfortunately, despite numerous studies demonstrating their value in estimating outcomes in MM, their widespread adoption has not yet occurred. Mounting evidence points to biomarker miRNAs as being a priority area for further research and validation.

From disease stage to drug resistance, miRNAs have been found to be predictive of patient outcomes. In a comparison of serum exosomal miRNAs in healthy patients, patients with smoldering MM, and patients with MM, distinct miRNA expression profiles were found to occur. Serum exosome derived levels of miR-20a-5p, miR103a-3p, and miR-4505 were different in healthy patients compared to those with smoldering MM or MM. Levels of let-7c-5p, miR-185-5p, and miR-4741 differed in MM patients versus those with smoldering MM and healthy patients [73]. Analysis of exosomal miRNA expression in 156 patients with MM compared with five healthy individuals found that lower expression of let-7b or miR-18a independently predicted inferior PFS and OS and complemented the mortality scoring tools of ISS and adverse cytogenics [74]. A study of 204 patients examining the miR profile of patients before and after drug resistance to Bortezomib, Thalidomide, and Lenalidomide found that downregulation of serum exosomal miR-16-5p, miR-15a-5p, miR-20a-5p, and miR-17-5p was associated with Bortezomib resistance [75]. In a meta-analysis of studies looking at the prognostic role of miRNAs in MM, 10 studies with a total number of 1214 patients were included and found that upregulated miR-92a and downregulated miR-16, miR-25, miR-15a, let-7e, and miR-19b were associated with poor prognosis in MM [76]. The United Kingdom Medical Research Council declared that based on the expression of just two miRNAs, miR-17 and miR-886-5p, MM patients could be risk stratified into three groups which would predict their prognosis better than the accepted gold standards of International Staging System/fluorescence hybridization and gene expression profiles [77]. Our own group recently found that high serum expression of miR-8074 was associated with a higher risk of progression-free survival shortening and a higher risk of overall survival shortening [78].

A recent meta-analysis of studies including more than 600 MM patients identified 37 differentially expressed miRNAs (DEMs). They found a statistically significant correlation between low expression of miR-30d-3p and reduced overall survival (OS) as well as high expression of miR-16-2-3p and reduced OS and progression-free survival (PFS) [79]. A 2017 study looked at seven miRNA datasets from MM patients and examined the number of predicted target genes that DEMs were associated with. Four miRNAs, miR-19a, miR-221, miR-25, and miR-223, were found to be associated with the highest number of genes and to be highly conservative in their sequences. This was suggested to be indicative of their role as prognostic or diagnostic biomarkers in MM [80]. A meta-analysis designed to examine the prognostic value of miRNA expression levels in MM concluded that downregulation of miRs-15a, 16, 25, 744, and let-7e was predictive of decreased OS and also that reduced PFS was predicted by reduced expression of miRs-15a, 16, 25, or increased expression of miR-92a [76]. In contrast, urinary expression of miRs-134-5p, 6500-5p, 548q, and 548y were found to be significantly reduced in MM patients in comparison to healthy controls. This pattern was preserved between newly diagnosed and relapsed groups and persisted in MM patients in remission. No significant prognostic data was demonstrated from urinary miRNA expression levels [81]. miR-720 and miR-1246 have been shown to be robust diagnostic biomarkers for MM and to be predictive of reduced PFS [82].

## 5. miRNA Based Therapies in MM

In light of the considerable impact of both oncomirs and tumor suppressor miRNAs on the pathogenesis and disease course of MM, miRNA-based therapies offer a unique approach from which to develop treatments for MM. Paracrine and endocrine effects of miRNAs are possible in MM through their release from MM cells in exosomes [83,84]. Exosomes have been used as vehicles in the delivery of miRNAs for various cancers in experimental settings [85]. Nanocarriers based on different synthetic and natural structures have been developed as vectors for miRNAs. Lipid-based carriers may be more effective at miRNA delivery if they include cholesterol or oleic acid in their formulations [86,87,88]. The most commonly used synthetic cationic polymer-based carriers are polyethylenimines [89]. Naturally derived cationic miRNA delivery vehicles frequently employ chitosan due to its established profile as a nanovector and its biocompatible and biodegradable properties [90]. Chitosan and PLGA have both been approved by regulatory bodies for use in humans and miR-34a-loaded chitosan/PLGA nanoplexes demonstrated cytotoxic effects in myeloma engrafted mice without any signs of toxicity [91]. Concerns over immunogenicity and the persistence of nanovectors with high surface charges led to the exploration of nanogels as an effective means of myeloma cell transfection with miR-34a [92].

Oncomirs may be targeted by means of miRNA sponges, which are RNA molecules with a complimentary amino-acid sequence to sequester the target miRNA [93]. A practical approach has been described for the construction of sophisticated miRNA sponges that can target multiple different miRNAs [94]. Oncomir knockdown has been achieved at the genetic level by means of technologies such as CRISPR/Cas9 and these methods may be applied to MM [95,96]. Peptide nucleic acids and locked nucleic acids (LNAs) can also be used to bind and decrease available oncomirs. Inhibition of miR-221 by means of an LNA construct was an effective means of reversing Melphalan resistance in a mouse model of MM [97]. In addition, miR-138 is a negative regulator of osteogenesis in MM and the effect of its overexpression can be reversed by using LNA-modified anti-miR-138 oligonucleotides in vitro and in vivo [98]. Related technology has been used to effectively manipulate let-7 target genes, including *MYC* in a mouse model of MM with a significant observed survival benefit and excellent putative applications outside of MM [99]. Clinical data on miRNA-based therapies in MM remains elusive as the only approved trial for this paradigm was halted early due to safety events in patients with other malignancies [100]. Preclinical areas of interest include the reversal of drug resistance to Melphalan and Bortezomib [97,101]. Increased MM cell sensitivity to Dexamethasone and Dexamethasone in combination with Doxorubicin and/or Bortezomib has also been demonstrated in preclinical settings by means of miRNA-based therapies [102].

## 6. Conclusions

miRNAs have a broad application prospect in MM diagnosis and treatment. Convincing studies have demonstrated that miRNA expression is dysregulated in MM, which affects maintenance of proliferative signaling, avoiding growth inhibitors, resistance to cell death, activation of invasion and induction of angiogenesis. Some miRNAs may be involved in multiple cancer-related signaling pathways, while others may be related to tumorigenesis by targeting oncogenes and tumor suppressor genes. Ongoing work in the area of miRNAs in MM demonstrates their undervalued role as diagnostic and prognostic biomarkers. Their role as therapeutics has been demonstrated preclinically and is best understood as sensitizing agents in combination with approved drugs. Combining miRNA therapeutics with chemotherapy may increase the anti-myeloma efficacy. More studies should be conducted to understand the mechanism completely and improve miRNA-based therapies in MM. Harnessing and disrupting the miRNA-based pathways in MM may be sufficient to arrest or reverse the progression of MM in the future.

## Figures and Tables

**Figure 1 jpm-12-01428-f001:**
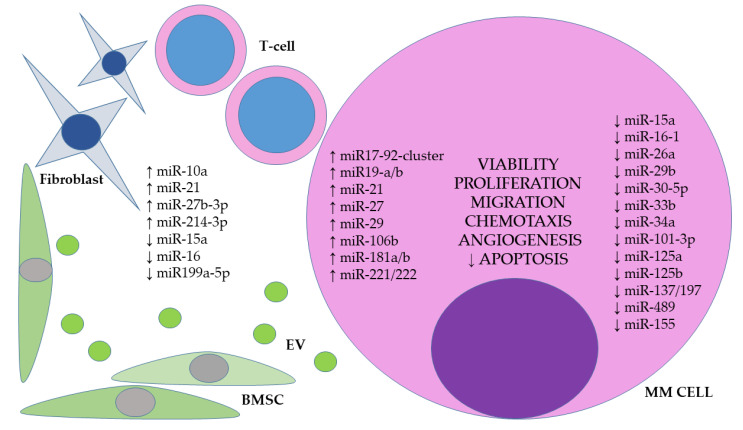
miRNA expression in MM pathogenesis. BMSC—bone marrow stem cells, EV—extracellular vesicle, MM–multiple myeloma, ↑—increased expression, ↓—decreased expression.

**Table 1 jpm-12-01428-t001:** List of selected oncomirs in MM. ↑—increased expression, ↓—decreased expression.

miRNA	Cell Processes	Targets	Reference
miR-10a	↑ proliferation ↓ apoptosis	EphA8, SEMA5A	[27,28,29]
miR-27b-3p miR-214-3p	↑ proliferation apoptosis resistance	FBXW7, PTEN/AKT/GSK3	[30]
miR-21	↓ T cells differentiation	STAT-1/-5a-5b STAT3	[31]
miR17-92 cluster	↑ proliferation ↓ apoptosis	SOCS1, BCL2	[37,38]
miR-21	↑ proliferation ↓ apoptosis	PTEN, BTG2, Rho-B	[39]
miR-221/222	↑ proliferation ↓ apoptosis	p27Kip1, p57Kip2, PTEN, PUMA	[40]
miR-181a/b	↓ apoptosis	PCAF, p53	[36,52]
miR-125a-5p miR-194-2/192 miR-215/194-1	↓ apoptosis	P53	[53,54]
miR-106b	↓ apoptosis	MAPK	[56]
miR-214-3p	↓ apoptosis	PTEN/AKT/GSK3 FBXW7	[30]
miR-19b/20a	↓ migration ↓ proliferation	PTEN	[57]
miR-27	↓ migration ↓ proliferation	SPRY2	[58]

**Table 2 jpm-12-01428-t002:** List of selected tumor suppressor miRNAs in MM. ↑—increased expression, ↓—decreased expression.

miRNA	Cell Processes	Targets	Reference
miR-15a/16	↓ tumor growth ↓ angiogenesis ↑ apoptosis	VEGF MAPK, AKT, NF-κB-activator MAP3KIP, S6 ribosomal protein	[32,59]
miR-199a-5p	↓ chemotaxis ↓ angiogenesis	VEGF, HIF-1α, IL-8, FGF-b	[33,34]
miR-29b	↑ apoptosis ↓ proliferation	SOCS1, PI3K/AKT, FOXP1	[43,44]
miR-26a	↓ migration ↓ proliferation ↑ apoptosis	CD38	[45]
miR-489	↓ proliferation ↓ viability	LDHA	[46]
miR-30-5p	↓ migration ↓ proliferation	BCL9	[47]
miR-34a	↓ proliferation ↑ apoptosis	NOTCH, MYC, BCL-2, CD44	[61,62,63]
miR-125a	↓ viability ↓ colony-forming ability	USP5	[65]
miR-125b	↓ tumor growth	IRF4	[66]
miR-33b	↓ viability ↓ migration ↓ colony-forming ability ↑ apoptosis	PIM-1	[67]
miR-155	↓ proliferation ↑ apoptosis	PSMβ5	[69]
miR-29b	↓ proliferation ↑ apoptosis	Sp1	[70]
miR-101-3p	↓ viability	BIRC5	[71]
miR-137/197	↓ viability ↓ migration ↓ colony-forming ability ↑ apoptosis	MCL-1	[72]

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
