# Peer review of "MiRNA as a Potential Target for Multiple Myeloma Therapy–Current Knowledge and Perspectives"

_jpm, 2022, doi:10.3390/jpm12091428_

Round 1
Reviewer 1 Report
Szudy-Szczyrek and colleagues provide an overview of the currently known miRNAs that are relevant in the context of multiple myeloma. Besides summarizing knowlegde on individual miRNAs, they also shed light on potential clinical implications.
Overall, this is a comprehensive and well-written review that warrants publication in the Journal of Personalized Medicine after a few suggested minor modifications:
1) The abstract appears very short and is particularly succinct on the topic of miRNAs. I would suggest adding more information on what to expect in the review to get readers interested.
2) Parts “1. Introduction“ and “2. MicroRNA“ contain a number of spelling and expression inconsistencies and need English language editing (e.g. l.41 “is“, l.72 “by...by“, l.76 “serve as“, l.76 “promising“)
3) Part 3.3. lines 196-199. This paragraph needs clarification and more details. MAPK and PI3K/Akt are pro-survival/anti-apoptotic pathways. What do the autors mean by “targeting“ these pathways? Ususally, this is meant in an inhibitory way which would, however , not increase resistance to apoptosis. I suspect this is just a matter of wording and I would suggest to add a few sentences to specify what is meant.
4) Part 6. Summary: I would prefer this paragraph to be titled “Conclusion“. Moreover, I would encourage the authors to write a separat chapter of half a page or more titled “summary“ which acutally provides a summarizing bracket that recapitulates all of the major points of the review in brief.
Author Response
Reviewer#1
Szudy-Szczyrek and colleagues provide an overview of the currently known miRNAs that are relevant in the context of multiple myeloma. Besides summarizing knowlegde on individual miRNAs, they also shed light on potential clinical implications.
Overall, this is a comprehensive and well-written review that warrants publication in the Journal of Personalized Medicine after a few suggested minor modifications:
1) The abstract appears very short and is particularly succinct on the topic of miRNAs. I would suggest adding more information on what to expect in the review to get readers interested.
Ad 1) Done as suggested.
2) Parts “1. Introduction“ and “2. MicroRNA“ contain a number of spelling and expression inconsistencies and need English language editing (e.g. l.41 “is“, l.72 “by...by“, l.76 “serve as“, l.76 “promising“)
Ad 2 ) Done as suggested.
3) Part 3.3. lines 196-199. This paragraph needs clarification and more details. MAPK and PI3K/Akt are pro-survival/anti-apoptotic pathways. What do the autors mean by “targeting“ these pathways? Ususally, this is meant in an inhibitory way which would, however , not increase resistance to apoptosis. I suspect this is just a matter of wording and I would suggest to add a few sentences to specify what is meant.
Ad 3) Thank you for the comment. We have clarified the mentioned paragraph in more details.
4) Part 6. Summary: I would prefer this paragraph to be titled “Conclusion“. Moreover, I would encourage the authors to write a separat chapter of half a page or more titled “summary“ which acutally provides a summarizing bracket that recapitulates all of the major points of the review in brief.
Ad 4) Done as suggested.
We would like to thank the Reviewer for both the review of the manuscript and for the suggestions.
Reviewer 2 Report
Dr Szudy-Szczyrek and Colleagues have provided a comprehensive review, focused on microRNAs (miRNAs) within the specific context of multiple myeloma (MM), revising the important functional role of miRNAs in both MM cells and the supportive bone marrow microenvironment. Specific criticisms are highlighted witin the following section.
1) Table 2. Second row: “miR-a/16”. Was it meant as “miR-15a/16”. Only reference #32 was included. This should be revised, in order to properly reference papers #56, where VEGF-related investigation was performed.
2) Authors have properly mentioned the prognostic role of serum-derived exosomes. The following studies should be added and referenced: Blood. 2017;129:2429-2436.
3) While talking about exosomes, It would be relevant to provide the Reader the existence of studies that have addressed the biological role of bone marrow-derived exosomes and the related miRNA content in supporting MM pathogenesis. The following studies should be added and referenced:
J Clin Invest. 2013;123:1542-1555;
- Cancer Lett. 2016 ;377:17-24.
4) Among studies that have addressed the relevance of miRNA in MM, the following studies are missing and should be added and referenced:
Blood Cancer J. 2019;9:21;
Leukemia. 2018;32:1739-1750;
Leukemia. 2017;31:853-860;
5) Throughout the paper, Authors sometimes refer to previously published studies without mentioning First Author’s name (these are the majority); sometimes First Author’s name is mentioned. Please, choose either or.
6) miR-155 has been discussed as a MM-oncomiR (Cancers, 2019). It would be relevant for the Reader to know that the oncogenic role of miR-155 has been reported for the first time in 2009, within the context of Waldenstrom Macroglobulinemia: Blood. 2009;113:4391-4402.
Author Response
Reviewer#2
Dr Szudy-Szczyrek and Colleagues have provided a comprehensive review, focused on microRNAs (miRNAs) within the specific context of multiple myeloma (MM), revising the important functional role of miRNAs in both MM cells and the supportive bone marrow microenvironment. Specific criticisms are highlighted witin the following section.
1) Table 2. Second row: “miR-a/16”. Was it meant as “miR-15a/16”. Only reference #32 was included. This should be revised, in order to properly reference papers #56, where VEGF-related investigation was performed.
Ad 1) We have corrected the table according to the suggestion.
2) Authors have properly mentioned the prognostic role of serum-derived exosomes. The following studies should be added and referenced: Blood. 2017;129:2429-2436.
Ad 2) We have added the mentioned data.
3) While talking about exosomes, It would be relevant to provide the Reader the existence of studies that have addressed the biological role of bone marrow-derived exosomes and the related miRNA content in supporting MM pathogenesis. The following studies should be added and referenced:
J Clin Invest. 2013;123:1542-1555;
- Cancer Lett. 2016 ;377:17-24.
Ad 3) We have added the mentioned data.
4) Among studies that have addressed the relevance of miRNA in MM, the following studies are missing and should be added and referenced:
Blood Cancer J. 2019;9:21;
Leukemia. 2018;32:1739-1750;
Leukemia. 2017;31:853-860;
Ad 4) We have added the mentioned data.
5) Throughout the paper, Authors sometimes refer to previously published studies without mentioning First Author’s name (these are the majority); sometimes First Author’s name is mentioned. Please, choose either or.
Ad 5) Done as suggested.
6) miR-155 has been discussed as a MM-oncomiR (Cancers, 2019). It would be relevant for the Reader to know that the oncogenic role of miR-155 has been reported for the first time in 2009, within the context of Waldenstrom Macroglobulinemia: Blood. 2009;113:4391-4402.
Ad 6) We have added the mentioned data.
We would like to thank the Reviewer for both the review of the manuscript and for the suggestions.
Round 2
Reviewer 2 Report
Authors have revised the manuscript according to the previously raised criticisms.